# Incremental Costs and Diners’ Satisfaction Associated with Improvement in Nutritional Value of Catering Dishes

**DOI:** 10.3390/nu14030617

**Published:** 2022-01-30

**Authors:** Ofira Katz-Shufan, Danit R. Shahar, Liron Sabag, Tzahit Simon-Tuval

**Affiliations:** 1Department of Public Health, The Health & Nutrition Innovative International Center, Faculty of Health Sciences, Ben-Gurion University of the Negev, P.O. Box 653, Beer-Sheva 8410501, Israel; dshahar@bgu.ac.il (D.R.S.); lironsharon1@gmail.com (L.S.); 2Department of Health Policy and Management, Guilford Glazer Faculty of Business and Management and Faculty of Health Sciences, Ben-Gurion University of the Negev, P.O. Box 653, Beer-Sheva 8410501, Israel; simont@bgu.ac.il

**Keywords:** catering service, diet quality, recipe modification, incremental cost, satisfaction

## Abstract

Eating in catering systems has been identified as a driver of poor diet quality. Interventions within catering systems increase the nutrient density of dishes. Little is known about the incremental costs associated with this strategy. One part of the NEKST (Nutrition Environmental Kibbutzim Study) intervention was nutritional improvement of recipes (decreasing the amount of energy, sodium, and saturated fat). We evaluated the nutritional content of dishes per 100 g and the incremental costs associated with these changes from the catering system’s perspective, as well as diners’ satisfaction with the catering system before and after the intervention. Our results revealed that as energy and saturated fat decreased, the associated incremental cost increased (rs = −0.593, *p* = 0.010 and rs = −0.748, *p* < 0.001, respectively). However, the decrease in sodium was not associated with increased costs (rs = 0.099, *p* = 0.696). While diners’ satisfaction decreased in the control group, it did not change in the intervention group following the intervention (*p* = 0.018). We concluded that recipe modification improved the nutritional value of dishes without increasing cost. This intervention was not associated with decreased diner satisfaction. This evidence encourages the implementation of policies to improve the nutritional quality of food served by caterers without jeopardizing sales and with the potential to improve public health.

## 1. Introduction

Chronic noncommunicable diseases (NCDs) pose a substantial economic burden on healthcare systems worldwide [1,2]. This burden is expected to escalate because of the increased prevalence of obesity, most notably in cardiovascular diseases, type 2 diabetes, and cancers [3]. Thus, intervention strategies targeted at adoption of healthy diets have the potential to result in economic benefits. Unhealthy dietary intake contains excess amounts of saturated fat (as in red and processed meats and dairy products) and sugar, as well as low intake of fruits and vegetables, whole grains, pulses, and nuts. This unhealthy diet has been associated with an increased risk of NCDs [4] and generates a larger burden than the combined burden of tobacco, alcohol, and physical inactivity [5].

Eating in a catering system has been identified as a driver of poor diet quality because of higher total energy intake, the energy contribution from fat, lower intake of micronutrients [6], and lower fruit and vegetable consumption [7]. In some cases, this was positively associated with body weight [8] and BMI [7]. On the other hand, catering systems may provide a promising environmental setting to promote healthy eating out [9] and enable a large group of regular diners to be addressed simultaneously. Multicomponent food polices that use existing structures and systems such as catering systems have appeared consistently to be an effective strategy to improve healthy eating [10,11]. This approach includes increasing the nutrient density of dishes provided by the catering system [12]. In order to enable caterers to successfully achieve this goal, food polices should strengthen caterers’ technical capacity, financial means, and human resources. In addition, such polices should not refer to healthy eating in general but rather should be tailored to local culture and preferences [9].

Little is known about the economic implications of intervention programs in catering systems [13,14]. The “Food Choice at Work” study implemented a complex dietary intervention in various manufacturing workplaces in Ireland. One component of the intervention was change in the food served, including dish modifications. As part of the evaluation of the intervention program, a cost analysis from the employer’s perspective [15] and a cost-effectiveness analysis from healthcare providers’ and employers’ perspectives [16] were conducted. Neither analysis addressed the incremental costs associated with modifying the nutrients of the food served.

In the current study, we show the results of an intervention study within a catering system, the NEKST (Nutrition Environmental Kibbutzim Study). The present study estimated the incremental costs associated with the nutritional change following recipe modification from the catering service’s perspective.

To the best of our knowledge, our study is the first to investigate the relationship between dietary changes in the dishes served by a catering service, in terms of changes in energy, macronutrients (saturated fat), and micronutrients (sodium), and the associated incremental costs.

In addition, the study addressed participants’ satisfaction, which is an important aspect when considering expected changes in sales following nutritional changes in the menu.

## 2. Materials and Methods

### 2.1. Study Setting and Population

The NEKST was an interventional study conducted in two catering systems of two kibbutzim (kibbutzes) in Israel, Kibbutz Magen (intervention group) and Kibbutz Nir-Yitzhak (control group), during a three-month period (March–June 2017). Participants were regular diners 30+ years of age who ate lunch at least three times a week in the kibbutz dining room and chose their food themselves (intervention *n* = 67 vs. control *n* = 67). Participants who agreed to participate in the study and met the inclusion criteria were recruited following their written informed consent. A detailed description of the study is presented elsewhere [17]. The NEKST intervention program included: nutritional improvement of recipes, environmental changes (changes in presentation locations of dishes and labeling of nutritional healthy dishes with a green “Like”), an ongoing healthy nutrition campaign in the dining room, and a nutrition and physical activity education program that was the single component administered in both intervention and control groups.

### 2.2. Recipe Modifications

As a part of the intervention program, nutritional changes in recipes were made according to dietary recommendations towards the Mediterranean diet [18,19], i.e., decreasing the amounts of processed foods, sugar, and salt; increasing the amounts of fruits, vegetables, and whole grains; and replacing other vegetable oils with canola and olive oils. The recipe modifications included: changing the type of meat and dairy products to lower-fat ones; using oat bran to replace breadcrumbs; adding vegetables; reducing the amounts of sodium-rich products such as salt, instant powdered soup, and grilled chicken powder; reducing the portions of industrial sauces (chili, pad-thai, teriyaki sauces); and reducing the amounts of sugar-rich products such as date syrup.

Information on recipe ingredients of lunch menu dishes and the actual serving frequency in the catering services of the intervention group was obtained from the catering service staff. We calculated the dishes’ nutrient content for each recipe using “Tzameret”, Nutrition Analysis Software [20]. Based on this information, we developed an algorithm for choosing recipes to modify during the intervention (Figure 1). The first step of the selection algorithm was based on the weekday serving frequency of the dishes and the number of dishes that were purchased by diners. Namely, we chose dishes that were served at least twice a month on weekdays and purchased on average by more than 20 diners per day. The second step was to determine whether a dish was less healthy or healthy according to the Israeli Ministry of Health recommendations [21]. Specifically, the less healthy dishes were selected for modification, namely, those with energy >300 Kcal per 100 g, or with saturated fat >5 g per 100 g, or with sodium >500 mg per 100 g.

### 2.3. Study Measures

#### 2.3.1. Nutrient Content of Dishes before and after Recipe Modification

We analyzed the amounts of energy, saturated fat, and sodium in a 100 g dish using “Tzameret” [20]. In this program, the inputs were all of the recipe ingredients, the preparation methods, and the number of portions produced by the recipe. The outputs were the nutritional values of the dish (energy, carbohydrates, protein, fat in general, types of fat, vitamins and minerals) per 100 g. The change in the nutritional value was calculated as the difference between the nutritional values per 100 g of the modified recipe and those of the original recipe. In this study, we present the changes in energy, saturated fat, and sodium, as those are central for risk reduction for NCDs [22].

#### 2.3.2. Incremental Costs Associated with Recipe Modification

This refers to the costs that were associated with the change in the ingredients used to prepare each dish. The costs were calculated from the catering system’s perspective, in other words, the actual costs paid by the caterer including taxes, subsidies, and discounts. The prices of each recipe’s ingredients were analyzed using Microsoft Office Excel (version 2013, Microsoft Corp., Redmond, WA, USA). These analyses included ingredients that were removed or reduced in the dishes, as well as those that were added or increased (e.g., in the Bolognese recipe, a dietitian recommended reducing the amount of instant powdered soup by half and replacing the tomato paste with a low-sodium one, as well as adding onions, carrots, and zucchini). In order to find the incremental cost per serving, the total recipe’s incremental cost was divided by the number of portions per recipe. For the calculation of the total incremental cost of purchased dishes, the incremental cost per serving was multiplied by the number of servings that were purchased during an arbitrarily selected three-week period during the intervention period (dates: 2–6 April, 23–27 April, 4–8 June 2017). Cost estimates and tariffs were obtained from the catering service accounting system in Israeli shekels (ILS) and converted to USD, assuming an exchange rate of 1 USD = 3.6 ILS. 

#### 2.3.3. Participant Satisfaction with the Catering System

This was assessed before the intervention (baseline) and at the completion of the intervention (after three months—Time 3) by five questions based on a questionnaire developed by Shahar et al. [23]. In this questionnaire, participants were asked to report their general satisfaction with the catering service and the food’s appearance, taste, variety, and healthiness on a 5-point scale ranging from 1 = very low to 5 = very high. An example question was “What is your level of satisfaction with the food served in the dining room in terms of the taste of the dishes served?” An aggregate satisfaction score was calculated as the average of the responses for the five statements. Cronbach’s alpha was calculated to check for coherence of the answers to the questions on satisfaction. Cronbach’s alpha of the satisfaction scores was 0.85 at baseline and 0.82 at Time 3. 

### 2.4. Statistical Analyses

The statistical analysis was carried out using IBM SPSS Statistics for Windows (version 23.0, IBM Corp., Armonk, NY, USA) Tests were considered significant at *p* < 0.05 (both sides). The Spearman correlation coefficient estimated the correlation between the nutritional change and the incremental costs associated with recipe modifications. To examine the impact of the intervention on between-group and within-group changes in diners’ satisfaction, we used a repeated measure ANOVA in which the core variable of interest was the interaction between the effects of time (before vs. after the intervention) and group (study vs. control). 

## 3. Results

### 3.1. Changes in Nutritional Value of Dishes following Recipe Modifications

Eighteen recipes were selected for modification. The nutrient changes to these recipes following the intervention are presented in Table 1. An analysis of the nutritional values after recipe changes revealed that the difference per 100 g in sodium levels ranged from a decrease of 486.23 mg (in Bolognese) to an increase of 7.69 mg (in cheese pie). The difference per 100 g in saturated fat ranged from a decrease of 4.22 g (in stuffed zucchini) to an increase of 0.13 g (in turkey wings). The difference in energy levels per 100 g ranged from a decrease of 215.55 Kcal (in beef schnitzel) to an increase of 1.09 Kcal (in turkey breast) (Table 1). Recipe changes led to a decrease in sodium in 14 (78%) of the modified recipes, in saturated fat in 11 (61%), and in energy in 14 (78%).

### 3.2. Incremental Costs Associated with Recipe Ingredients Modifications

Table 2 presents the incremental costs associated with recipe ingredient modification. Increases in the costs associated with these modifications mainly derived from changing the type of meat and dairy products, adding oat bran instead of breadcrumbs, and adding vegetables. Reductions in these costs mainly derived from reductions in high-sodium ingredients such as salt, instant powdered soup, and instant grilled chicken powder; reductions in industrial sauces added; using cooking ingredients with lower fat content; and reductions in sugary products such as date syrup. The incremental costs associated with recipe changes ranged between $−0.06 (turkey wings) and $+0.51 (meatballs) per serving. The change in recipes led to a positive (greater than zero) incremental cost in nine recipes (50% of those modified). Naturally, the aggregate incremental costs associated with a recipe’s modification depended on the frequency at which each dish was actually purchased. The total number of servings that underwent recipe modification and were purchased during an arbitrarily selected three-week period throughout the intervention was 827. The total incremental cost associated with modification of recipes purchased was $90.99. The mean incremental cost per serving was $0.11.

### 3.3. The Association between Nutrient Changes and Incremental Costs

Analysis of the association between nutrient changes per 100 g and the incremental costs per serving incurred by recipe modification revealed that as the saturated fat decreased, the incremental cost associated with this modification increased (r_s_ = −0.748, *p* < 0.001). A similar finding was found with regard to the energy value of the recipes (r_s_= −0.593, *p* = 0.010). However, decreases in sodium were not associated with increased costs (r_s_ = 0.099, *p* = 0.696).

### 3.4. Change in Satisfaction with the Catering Service

The analysis included 47 participants in the intervention group and 40 participants in the control group. Both groups were comparable at baseline regarding age (*p* = 0.520), gender (*p* = 0.446), educational level (*p* = 0.184), and BMI (*p* = 0.506) (Table 3). Table 4 presents the mean (±SE) diners’ satisfaction with the catering service before and after the intervention. While there was a decrease in the average satisfaction with the catering service among the control group, no change was observed in the intervention group following the program (*p* = 0.018). The significant between-group differences following the intervention stemmed predominantly from the general level of satisfaction with the kitchen and dining room (*p* = 0.037), the variety of dishes served (*p* = 0.007), and the nutritional values of the dishes served (*p* = 0.045).

## 4. Discussion

This study evaluated changes in energy, sodium, and saturated fat in dishes, the associated incremental costs, and diner satisfaction following the NEKST intervention program. We found that nutritional improvements to dishes were made without increase in cost or decrease in diners’ satisfaction. Specifically, while the incremental costs increased when the energy and saturated fat content of the dishes decreased, it did not increase when the sodium content of the dishes decreased. In addition, no decrease in diner satisfaction was observed following the NEKST intervention. 

To the best of our knowledge, our study is the first to test both the nutritional changes and the incremental costs associated with recipe modification. For example, in the DIRECT study [24], which took place in a workplace in Israel, recipes of dishes served in the dining room for lunch were modified. In this case, the change was in accord with the three diets that were tested, namely, low-fat, low-carbohydrate, and Mediterranean diets. Cost analysis was not performed. The “Food Choice at Work” group conducted a cost analysis [15] and a cost-effectiveness analysis [16] of an integrated intervention program that included menu modifications, but it did not calculate the component costs. 

The modification of recipes on the menu in the intervention catering led to more healthy nutritional values of offered dishes, with decreased energy, saturated fat, and sodium. Reduced consumption of saturated fat leads to risk reduction for cardiovascular disease, diabetes, and colon cancer [25]. Lower consumption of energy reduces the risk of overweightness and obesity, which are correlated with risk reduction for hypertension, cardiovascular disease, and cancer [25]. High sodium levels in foods contribute to higher risk of hypertension and cardiovascular disease [26]. Hence, reducing these nutrients in dishes may be an effective strategy to improve public health. In our study, the decrease in sodium levels was not associated with an increase in incremental costs. Moreover, the total increase in cost per serving was rather moderate, from the catering system perspective. This may demonstrate that it is possible for caterers to adopt this intervention strategy without facing a financial burden [27]. In addition, although the total incremental cost associated with recipe modification was positive (greater than zero), we believe that it may be relatively modest compared with the potential savings associated with the reduced risks of NCDs.

An environmental approach targeted at the adoption of healthy eating habits overrides the psychological pressure associated with individual-based approaches [28]. Recipe modification is one example of this approach. One advantage of recipe modification is that it encourages a healthy diet through creation of an enabling environment that supports responsible behavior by changing caterers’ practices and creating healthier defaults [28]. Because recipe modifications may negatively alter the taste and appearance of a dish, it may cause diners to avoid purchasing it. There is often a concern that healthier food is less palatable [29]; thus, it may reduce diner satisfaction. In our study, while there was a decrease in diners’ satisfaction in the control group, satisfaction with the catering service after the intervention did not change in the intervention group. This result may indicate that the intervention changes were accepted by diners and may not have deterred them from choosing the modified dishes. This finding is similar to those from other intervention studies in which no change in satisfaction was observed after offering healthy meals [30,31]. In several studies, there was even a positive change in diner satisfaction following changes in food served towards healthier options [32,33,34,35]. The increase in satisfaction with the catering system in these studies was due to the taste of the food [32,33], the food presentation [35], the variety of dishes offered [32,33,34], and the availability of healthy dishes [32,33,34,35]. Thus, it seems that catering services can adopt similar food policies without compromising their sales.

Our study has three limitations. First, the purposive selection of the catering service limits the generalization of our results. However, this limitation does not weaken our conclusions, since our objective was not to provide cost estimates but rather to explore the association of these estimates with the corresponding nutritional improvements. Second, neither of the study groups is representative because of our low sample size and low response rates. Further large-scale studies are warranted to substantiate study results. Third, during the study, we encountered a change in the management of the catering service in the control group. This change may have influenced the control diners’ satisfaction (which decreased at Time 3). Thus, one may address the between-group comparison of diner satisfaction with caution yet still address the within-group comparison as valid.

## 5. Conclusions

The objective of our study was to provide a unique and actual estimate of the nutritional changes following recipe modification, the change in diners’ satisfaction, and the incremental costs associated with recipe modification from the catering service’s perspective. We conclude that, while decreased energy and saturated fat were associated with increased incremental costs, decreased sodium was not. In addition, nutritional improvements in the catering dishes were not followed by a decrease in diners’ satisfaction. Our results enable evidence-based decision making for caterers when facing the dilemma of whether to adopt nutritional changes in their menus and adhere to the implementation of food policies without jeopardizing their profitability. Our results may set a crucial basis for future cost-effectiveness analysis of similar strategies. Further studies that analyze modification of different recipes in other settings are warranted in order to substantiate our results. These may be targeted at more diverse and larger populations. This body of research could contribute to development of catering-level interventions and provide worthy evidence for food environment policymaking.

## Figures and Tables

**Figure 1 nutrients-14-00617-f001:**
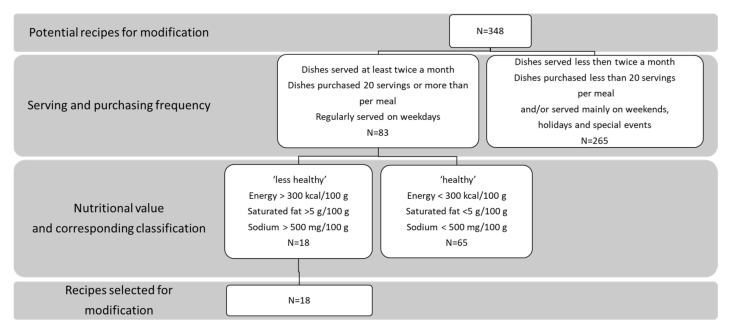
Algorithm for choosing recipes for modification.

**Table 1 nutrients-14-00617-t001:** Nutrient change per 100 g following recipe modification.

Recipe	Energy (Kcal)	Saturated Fat (g)	Sodium (mg)
Beef schnitzel	−215.55	−3.78	−30.45
Bolognese	−29.96	−0.66	−486.23
Cannelloni	−55.43	−3.88	4.01
Cheese pastry	−10.18	−0.78	3.28
Cheese pie	−23.86	−1.82	7.69
Chicken breast	0.09	0.03	−223.41
Chicken cutlets	−32.15	−0.35	−264.44
Chicken liver	0.96	0.01	−141.89
Chicken skewers	−0.08	0.01	−78.51
Meatballs	−25.01	−1.11	−288.91
Meatloaf	−12.73	−0.28	−206.61
Shepherd’s pie	−13.06	−0.3	−393.87
Spicy carrot salad	−0.39	0.02	−387.65
Stir-fried chicken	−8.18	−0.09	−116.21
Stir-fried tofu	−0.62	0	−55.69
Stuffed zucchini	−60.66	−4.22	7.47
Turkey breast	1.09	0.04	−146.34
Turkey wings	0.23	0.13	−174.02

**Table 2 nutrients-14-00617-t002:** Incremental costs (USD) associated with recipe ingredients modifications.

Recipe Name	Number of Servings Per Recipe	Incremental Cost Per Recipe ($)	Incremental Cost Per Serving ($)	No. of Servings Purchased *	Sum Incremental Cost of Purchased Recipe ($)
Beef schnitzel	120	−2.2	−0.02	54	−1.08
Bolognese	450	84.0	0.19	95	18.05
Cannelloni	60	9.4	0.16	NS	NS
Cheese pastry	25	11.5	0.46	NS	NS
Cheese pie	50	11.5	0.23	56	12.88
Chicken breast	150	−7.7	−0.05	117	−5.85
Chicken cutlets	200	0.9	0.01	NS	NS
Chicken liver	70	0.0	0.00	NS	NS
Chicken skewers	130	−3.6	−0.03	85	−2.55
Meatballs	400	203.8	0.51	125	63.75
Meatloaf	80	2.5	0.03	25	0.75
Shepherd’s pie	72	2.2	0.03	95	2.85
Spicy carrot salad	25	−0.1	−0.01	NS	NS
Stir-fried chicken	75	−2.6	−0.03	40	−1.2
Stir-fried tofu	25	−0.8	−0.03	37	−1.11
Stuffed zucchini	20	9.4	0.47	12	5.64
Turkey breast	300	−2.5	−0.01	86	−0.86
Turkey wings	77	−4.9	−0.06	NS	NS
For servings purchased (3 weeks)				827	90.99
For servings purchased (12 weeks)—estimated				3308	363.96

NS = Not served during the arbitrarily selected three weeks following the intervention. * The number of servings that were purchased during the arbitrarily selected three weeks during the intervention.

**Table 3 nutrients-14-00617-t003:** Baseline characteristics of the participants by study group.

	Intervention Group (Magen)	Control Group (Nir Yitzhak)	*p*-Value between Groups
*n*	47	40	
Age (years) ^a^	56.8 ± 15.6	60.3 ± 14.9	0.520 ^b^
Gender—male ^c^	22 (46.8%)	22 (55.0%)	0.446 ^d^
Educational level ^c^ Academic education	29 (61.7%)	19 (47.5%)	0.184 ^d^
BMI (kg/m^2^) ^a^	28.3 ± 5.0	27.4 ± 4.7	0.506 ^b^

^a^ Values are mean ± SD. ^b^ Mann–Whitney rank sum test. ^c^ Values are n (%). ^d^ χ^2^.

**Table 4 nutrients-14-00617-t004:** Changes in diner satisfaction with the catering service following the intervention.

How Satisfied Are You With	Intervention Group (Magen) *n* = 47		Control Group (Nir Yitzhak) *n* = 40		*p*-Value ^b^
	Before ^a^	After ^a^	Before ^a^	After ^a^	
Kitchen and dining room	2.81 ± 0.14	2.92 ± 0.11	3.20 ± 0.16	2.95 ± 0.12	0.037
Appearance of the dishes	2.81 ± 0.11	2.85 ± 0.10	3.03 ± 0.12	2.80 ± 0.11	0.112
Taste of the dishes	2.87 ± 0.13	2.75 ± 0.10	3.05 ± 0.14	2.90 ± 0.11	0.893
Variety of dishes	3.23 ± 0.12	3.17 ± 0.12	3.50 ± 0.12	2.95 ± 0.13	0.007
Nutritional value of the dishes	2.36 ± 0.11	2.49 ± 0.11	3.03 ± 0.12	2.83 ± 0.12	0.045
Average satisfaction with the catering system	2.82 ± 0.10	2.83 ± 0.08	3.16 ± 0.11	2.89 ± 0.09	0.018

^a^ Values are mean ± SE; ^b^ Repeated measure ANOVA, the interaction between time (before vs. after the intervention) and groups (study vs. control).

## Data Availability

O.K.-S. can provide all original data for review.

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
