# Peer review of "Incremental Costs and Diners’ Satisfaction Associated with Improvement in Nutritional Value of Catering Dishes"

_nutrients, 2022, doi:10.3390/nu14030617_

Round 1

Reviewer 1 Report

Dear Authors, Dear Editor,

This is a very interesting article that provides an analysis of the incremental costs and changes in satisfaction of lunch eaters in the catering system as a result of the improved nutritional value of the food. The latter aspect of the analysis should be included in the title. The authors demonstrated that nutritional interventions in catering systems can improve public health and, in addition, do not dramatically increase incremental food costs. Interestingly, customers rated the taste of the food minimally lower when the salt content was significantly reduced, which would imply that the food was over-salted.

The paper is well developed and structured and well justified in relation to the findings of previous research, both in the introduction and in the discussion. But still, the authors should provide further information about the study participants, e.g. gender, age (we only know that they were aged 30+), occupational status etc.  It should also be explained why the number of participants in the satisfaction survey (subsection 3.4.) was lower in both groups than the number N = 67 indicated in subsection 2.1.

And a few more minor comments:

L. 35 - also dairy products provide large amounts of saturated fat.

L. 36 - current so-called western diets are also characterised by a low intake of pulses

L. 39 - I propose to start the sentence with “Eating in a catering system”; one cannot speak of a poor diet in relation to the whole market for eating out, because the market also includes segments where the food offer is good or very good (e.g. casual dining, fine dining, premium or top premium, vegetarian or vege restaurants etc.).

L. 53 and 218 - in this article I would like to know a little more about the "Food Choice at Work" group (country, workers in what professions/workplaces, what the intervention was about).

L. 63-64 - use singular: macronutrient (energy is not a nutrient) and micronutrient (sodium only).

L. 68 - in this subsection it should be completed in which period the nutritional intervention was conducted (as precisely as it is stated in which days the costing was carried out).

L. 130 - the sentence needs to be corrected as it implies that the satisfaction assessment was carried out three months after the end of the intervention (I think this was not the case?).

L. 153, 155-156, 168-169 - names of types of food can be written in lower case.

L. 176 - for what reasons were the few dishes listed in Table 2 not served?

L. 205 - in the header of Table 4, the letters “b, c, d” should be written in superscript and explained below the table.

L. 215 - please write a little more about the DIRECT study (country, workers in which professions/workplaces, what the intervention consisted of).

L. 227 - the word 'high' was omitted before “blood pressure”; the term 'hypertension' could also be used.

L. 241 - I suggest referring to several literature sources for evidence that "There is often a concern that healthier food is less palatable".

L. 252 - in my opinion, the small size of the intervention and control samples was also a limitation of the study.

Given the limitations identified, the authors should identify the opportunities for future research in line with the theoretical and practical implications derived from the study.

Kind regards.

Author Response

Manuscript ID: nutrients-1552762

Incremental costs associated with improvement in nutritional value of catering dishes

Responses to Reviewer #1:                                                                                                           We would like to thank the reviewer for his/her helpful comments. Our responses are listed below. For convenience, we numbered the reviewer’s comments.

  1. This is a very interesting article that provides an analysis of the incremental costs and changes in satisfaction of lunch eaters in the catering system as a result of the improved nutritional value of the food. The latter aspect of the analysis should be included in the title. The authors demonstrated that nutritional interventions in catering systems can improve public health and, in addition, do not dramatically increase incremental food costs. Interestingly, customers rated the taste of the food minimally lower when the salt content was significantly reduced, which would imply that the food was over-salted.

Response: Thank you. Following the reviewer’s comment, we revised the title. It now reads: Incremental costs and diners' satisfaction associated with improvement in nutritional value of catering dishes.

  1. The paper is well developed and structured and well justified in relation to the findings of previous research, both in the introduction and in the discussion. But still, the authors should provide further information about the study participants, e.g. gender, age (we only know that they were aged 30+), occupational status etc.  It should also be explained why the number of participants in the satisfaction survey (subsection 3.4.) was lower in both groups than the number N = 67 indicated in subsection 2.1.

Response: Indeed, age 30 + was one of the study's inclusion criteria (as appears in the Methods section). A more detailed description of the study’s participants is provided in Table 3 and in the text regarding age, gender, education and BMI (please see: lines 199-200).

The number of participants that were included in the analyses of change in diners’ satisfaction (subsection 3.4) was lower than the total n of our study population (indicated in subsection 2.1), because not all participants completed this specific part of the questionnaire before and after the intervention.

And a few more minor comments:

  1. 35 - also dairy products provide large amounts of saturated fat.

Response: We agree with the reviewer and added this information accordingly (please see line 37).

  1. 36 - current so-called western diets are also characterized by a low intake of pulses

Response: We agree and added pulses to the sentence (please see line 38).

  1. 39 - I propose to start the sentence with “Eating in a catering system”; one cannot speak of a poor diet in relation to the whole market for eating out, because the market also includes segments where the food offer is good or very good (e.g. casual dining, fine dining, premium or top premium, vegetarian or veg restaurants etc.).

Response: Thank you. The sentence was revised as suggested (please see line 41).

  1. 53 and 218 - in this article I would like to know a little more about the "Food Choice at Work" group (country, workers in what professions/workplaces, what the intervention was about).
  2. 63-64 - use singular: macronutrient (energy is not a nutrient) and micronutrient (sodium only).

Response: We agree and revised the phrasing accordingly throughout the manuscript. It now reads: "To the best of our knowledge, our study is the first to investigate the relationship between the dietary changes in the served dishes by a catering service in terms of change in energy, macronutrient (saturated fat) and micronutrient (sodium) and the associated incremental costs." (Line 66-69). And: "This study evaluated the changes of energy, sodium and saturated fat in dishes," (Line 219).

  1. intervention was conducted (as precisely as it is stated in which days the costing was carried out).

Response: Thank you. We added the exact period (March-June, 2017) to the text (please see line 80).

  1. 130 - the sentence needs to be corrected as it implies that the satisfaction assessment was carried out three months after the end of the intervention (I think this was not the case?).

Response: Thank you. We revised the sentence. It now reads: "This was assessed before the intervention (baseline) and at the completion of the intervention (after three months -Time 3)." (please see line 141-142).

  1. 153, 155-156, 168-169 - names of types of food can be written in lower case.

Response: Thank you. This was revised accordingly throughout the text.

  1. 176 - for what reasons were the few dishes listed in Table 2 not served?

Response: Some of the dishes that were modified were not served during the arbitrarily selected three-week period of data collection. It is important to note that the decision on the weekly menu is made by the catering services staff and may be based on random, not systematic considerations such as availability of ingredients, and other preferences of the kitchen stuff.

  1. 205 - in the header of Table 4, the letters “b, c, d” should be written in superscript and explained below the table.

Response: Thank you. This was an editing mistake. The superscript was corrected, and the Table legend was added.

  1. 215 - please write a little more about the DIRECT study (country, workers in which professions/workplaces, what the intervention consisted of).

Response: We added more information about this study. It now reads:" For example, in the DIRECT study[23], that took place in a workplace in Israel, recipes of dishes served in the dining room for lunch were modified. In this case the change was in accord with the 3 diets that were tested, low fat, low carbohydrates and Mediterranean diet. Cost analysis was not performed." (line 229-232).

  1. 227 - the word 'high' was omitted before “blood pressure”; the term 'hypertension' could also be used.

Response: Thank you. We revised the wording to hypertension.

  1. 241 - I suggest referring to several literature sources for evidence that "There is often a concern that healthier food is less palatable".

Response: Thank you. We added ref [29] as the source for this concern.

  1. 252 - in my opinion, the small size of the intervention and control samples was also a limitation of the study.

Response: We agree and added the small sample size as an additional limitation (please see line 273-275).

  1. Given the limitations identified, the authors should identify the opportunities for future research in line with the theoretical and practical implications derived from the study.

Response: We agree. We revised the text accordingly. It now reads: "Further studies that analyze modification of different recipes in other settings are warranted in order to substantiate our results. These may be targeted at a more diverse and large populations. This body of research could contribute to development of catering-level interventions and provide worthy evidence for food environment policymaking."

Reviewer 2 Report

Shufan and colleagues address in their paper the important topic of cost associated with health-promoting eating. For their study, they chose the catering environment, where systematic changes of the nutritional value of applied foods are more likely to show higher impact on public health. They claim that, after adapting 18 of the originally applied 348 recipes, they achieved a diet that contained significantly less sugar, saturated fat, and salt, being therefore “healthier”, though at a moderately higher cost.

This is a routinely written study, in flawless English, well structured and easy to understand. However, the authors should mandatorily use another statistical evaluation method (repeated measures ANOVA with Tukey’s post-hoc test).

The result section would largely benefit from a table informing about how much energy, salt, and SFA (and possibly also other nutrients such as sugar) per average meal were finally saved during the intervention. This seems of particular relevance as they only changed 18 of the originally applied 348 recipes.

Moreover, they should be more careful when raising some health claims coming from their diet as they did not include the amounts of foods consumed and therefore cannot claim that the diners consumed less energy, saturated fat, or salt.

Abstract:

Line 22-23: The informative value of the abstract can be improved, as, given the significantly negative correlations between the content of energy and saturated fat on the one hand and cost on the other, the statement “… that nutritional improvements of dishes were not always associated with increased incremental costs …” is confusing. Instead, the best possible constructive instruction on how improve the nutritional value without increasing cost would be more appropriate here.

Introduction

Line 32: please write “type 2 diabetes” instead of just “diabetes”

Methods

Lines 82-90: The authors should make clear to which extent they accomplished the suggested dietary changes as suggested in References # 18 and 19 and should also make clear which of the suggested changes were not feasible within their experimental setting.

Line 102: I guess the authors mean here “…with saturated fat >5 g per 100 g,…” and not “…with saturated fat >5 mg per 100 g,…”.

Line 109-111: Seemingly, the authors have also evaluated the content of other nutrients, namely micronutrients, of the meals before and after. To substantiate their claim that the modified meals were “healthier”, it would be useful to also obtain some information on the changes of the other nutrients.

Line 115, 116 and other: I believe that “cost” is a term that, comparably to e.g. “snow”, is used in singular form only.

Line 127: Please add the years to the dates.

Line 138: It should be made clear that Cronbach’s alpha was calculated to check for coherence of the answers to the questions on satisfaction and a reference for the applicability of Cronbach’s alpha should be added.

Statistics

Lines 141-147: I don’t understand why the authors didn’t apply a repeated-measures ANOVA instead of using two-group comparisons such as the Mann-Whitney-U test and Wilcoxon test. They are missing out the information on the interaction term (which they later have difficulties to explain, getting to the differences between the verum and the control group). I do consider a re-evaluation using repeated-measures ANOVA as mandatory, with Tukey’s post-hoc test.

Results

Lines 150-158: It is difficult to understand why in some of the dishes the contents of salt, sugar and SFAs increased at all, despite the targeted replacement of foods having high contents of these compounds.

Line 165-166: The authors state that the changes of cost amongst other resulted from the reduction of “…instant powdered soup, and instant grilled 165 chicken powder; a reduction in industrial sauces added…”. Avoiding the usage of highly processed foods as the ones mentioned it typically associated with more effort and therefore higher labour cost. Did the authors consider these?

Line 181: Replace “serving which” with “serving, which”.

The legend of Table 4 should indicate to which statistical test the p values apply.

Discussion

Line 207: The text “This study evaluated the nutrient changes in dishes…” should be replaced with “This study evaluated the changes of three nutrients in dishes…” as the selected nutrients are far from representing an analysis of nutritional adequacy.

Lines 218-218: I believe that the “16” in “cost-effectiveness analysis16” should indicate a reference, but the brackets are missing here.

Line 224-228. The authors do not possess information on whether the diners consumed more or less energy as they cannot exclude the possibility that the diners ate more of the calorie-reduced meals. Please rephrase this sentence and, if citing a study here that supports the claim, please use one that has proven that the ad-libitum consumption of meals with a lower energy density had the desired health effects.

In general, I believe that the Discussion should also point out that the increase in cost per meal were rather moderate.

Author Response

Manuscript ID: nutrients-1552762

Incremental costs associated with improvement in nutritional value of catering dishes

Responses to Reviewer #2:                                                                                                           We would like to thank the reviewer for his/her helpful comments. Our response listed below. For convenience we numbered the reviewer comments.

  1. Shufan and colleagues address in their paper the important topic of cost associated with health-promoting eating. For their study, they chose the catering environment, where systematic changes of the nutritional value of applied foods are more likely to show higher impact on public health. They claim that, after adapting 18 of the originally applied 348 recipes, they achieved a diet that contained significantly less sugar, saturated fat, and salt, being therefore “healthier”, though at a moderately higher cost.

This is a routinely written study, in flawless English, well structured and easy to understand. However, the authors should mandatorily use another statistical evaluation method (repeated measures ANOVA with Tukey’s post-hoc test).

Response: Based on the reviewer comment we tested an interaction term. Significant interaction between the effects of time (before vs. after the intervention) and groups (study vs. control) was detected (F(1,85) = 5.801, p = 0.018). Our analyses, presented in table 4 provide more detailed description of the change in satisfaction observed in both study groups and with respect to the taste, variety and nutritional values of the dishes served.

  1. The result section would largely benefit from a table informing about how much energy, salt, and SFA (and possibly also other nutrients such as sugar) per average meal were finally saved during the intervention. This seems of particular relevance as they only changed 18 of the originally applied 348 recipes.

Response: Table 1 presents the changes in energy, saturated fat and sodium in each of the modified recipes per 100 grams. Since each meal consists of different food items, it was not feasible to estimate the change per meal.

The 18 recipes selected for modification were selected as most suitable for modification following the algorithm we developed and presented in Figure 1. Hence, these are those with the potential to greatly affect the overall nutritional value of the food served.

  1. Moreover, they should be more careful when raising some health claims coming from their diet as they did not include the amounts of foods consumed and therefore cannot claim that the diners consumed less energy, saturated fat, or salt.

Response: We agree with the reviewer. We changed the phrasing in the Discussion. It now reads: “Hence, reducing these nutrients in dishes may be an effective strategy to improve public health” (please see line 243-245).

  1. Abstract: Line 22-23: The informative value of the abstract can be improved, as, given the significantly negative correlations between the content of energy and saturated fat on the one hand and cost on the other, the statement “… that nutritional improvements of dishes were not always associated with increased incremental costs …” is confusing. Instead, the best possible constructive instruction on how improve the nutritional value without increasing cost would be more appropriate here.

Response: We agree with the reviewer and revised the phrasing. It now reads: “We conclude that recipes modification improve the nutritional value of dishes without increasing cost.It was not associated with decreased diners’ satisfaction."

Following this comment we revised similar statement in the Discussion. It now reads: "We found that nutritional improvements of dishes were improved without increase in cost and decrease in diners’ satisfaction."

  1. Introduction: Line 32: please write “type 2 diabetes” instead of just “diabetes”

 Response: Done.

Methods

  1. Lines 82-90: The authors should make clear to which extent they accomplished the suggested dietary changes as suggested in References # 18 and 19 and should also make clear which of the suggested changes were not feasible within their experimental setting.

 Response: The change of recipes was done following the rationale of the Mediterranean diet, using key ingredients in the Mediterranean diet, all mentioned in the text (lines 92-95). The components of the Mediterranean diet that were not included in our study recipes' modifications were fish and pulses, as such components were not included among the 18 recipes selected by the algorithm presented in Figure 1.

  1. Line 102: I guess the authors mean here “…with saturated fat >5 g per 100 g,…” and not “…with saturated fat >5 mg per 100 g,…”.

Response: Thank you, we corrected it.

  1. Line 109-111: Seemingly, the authors have also evaluated the content of other nutrients, namely micronutrients, of the meals before and after. To substantiate their claim that the modified meals were “healthier”, it would be useful to also obtain some information on the changes of the other nutrients.

Response: Indeed we evaluated the change in additional nutrients following the recipes modification, however, in the current study we focused on the three that are mostly associated with reduced risk of NCDs. We revised the text to address this. It now reads: “In this study, we present the changes in energy, saturated fat and sodium, as those are central for risk reduction of NCDs [22].” (line 122-123).

In addition, we wanted to analyze sugar content as well, but unfortunately there was not enough data from food suppliers about the sugar content (apart from the sum of carbohydrate) in recipes’ ingredients. From 2019 the Israeli Nutritional labeling was changed to have this important data.

  1. Line 115, 116 and other: I believe that “cost” is a term that, comparably to e.g. “snow”, is used in singular form only.

Response: Following consultation with our language editor, we believe that the words “cost” and “costs” were used correctly throughout the manuscript.

  1. Line 127: Please add the years to the dates.

Response: Done.

  1. Line 138: It should be made clear that Cronbach’s alpha was calculated to check for coherence of the answers to the questions on satisfaction and a reference for the applicability of Cronbach’s alpha should be added.

Response: Thank you. The sentence was revised (Please see lines 148-149).

  1. Statistics: Lines 141-147: I don’t understand why the authors didn’t apply a repeated-measures ANOVA instead of using two-group comparisons such as the Mann-Whitney-U test and Wilcoxon test. They are missing out the information on the interaction term (which they later have difficulties to explain, getting to the differences between the verum and the control group). I do consider a re-evaluation using repeated-measures ANOVA as mandatory, with Tukey’s post-hoc test.

Response: Please see our response to comment#1.

Results

  1. Lines 150-158: It is difficult to understand why in some of the dishes the contents of salt, sugar and SFAs increased at all, despite the targeted replacement of foods having high contents of these compounds.

Response: The explanation derives from the fact that when we replaced an ingredient in a recipe, that contained different amounts of various nutrients, it may lead to an increase in another undesired nutrient. For example, the replacement of high to low percentage of fat in the Cannelloni recipe resulted in modest increase in sodium.  Thus, the change in the cannelloni recipe led to a reduction in calories and saturated fat jointly with a small increase in the amount of sodium.Following a consultation with the study dietitian, the decision to modify recipe was made only in cases where the increase in undesired nutrient was significantly smaller than the decrease in the other undesired nutrients.

  1. Line 165-166: The authors state that the changes of cost amongst other resulted from the reduction of “…instant powdered soup, and instant grilled 165 chicken powder; a reduction in industrial sauces added…”. Avoiding the usage of highly processed foods as the ones mentioned it typically associated with more effort and therefore higher labour cost. Did the authors consider these?

Response: We agree with the reviewer. During our study we consulted with the catering system management and staff to inquire whether the change in recipes also involve a change in working hours, and in the amount of electricity, gas and water consumed. It turned out that from the catering system's perspective, these were negligible thus were not included in the analyses.

  1. Line 181: Replace “serving which” with “serving, which”.

Response: Thank you, was changed.

  1. The legend of Table 4 should indicate to which statistical test the p values apply.

Response: This was an editing mistake. The legend of the Table and the footnotes were added.

Discussion

  1. Line 207: The text “This study evaluated the nutrient changes in dishes…” should be replaced with “This study evaluated the changes of three nutrients in dishes…” as the selected nutrients are far from representing an analysis of nutritional adequacy.

Response: We agree. The sentence was revised. It now reads: “This study evaluated the changes of energy, sodium and saturated fat in dishes,…” (Please see line 219).

  1. Lines 218-218: I believe that the “16” in “cost-effectiveness analysis16” should indicate a reference, but the brackets are missing here.

Response: Thank you. It is now corrected.

  1. Line 224-228. The authors do not possess information on whether the diners consumed more or less energy as they cannot exclude the possibility that the diners ate more of the calorie-reduced meals. Please rephrase this sentence and, if citing a study here that supports the claim, please use one that has proven that the ad-libitum consumption of meals with a lower energy density had the desired health effects.

Response: We agree. However, a key component in nutrition-enhancing strategies to promote public health is the reduction of undesired nutrients in food servings, regardless of the amount consumed. We revised the sentence to address this fact (please see lines 244)

  1. In general, I believe that the Discussion should also point out that the increase in cost per meal were rather moderate.

Response: We agree. We added to the text the following sentence: “Moreover the total increase in cost per serving was rather moderate, from the catering system perspective.” (Please see lines 246-247).

Round 2

Reviewer 2 Report

Shufan and colleagues have made amendments to their original manuscript on cost associated with health-promoting eating. The quality of the paper has largely improved, but two points, one major and one minor, still demand to be addressed before the manuscript finally can be recommended for publication.

The authors’ response to Points # 1 and #12 (as numbered by the authors) is confusing and contradictory. In Lines 151-158, it is still written that they applied two-group comparisons such as the Mann-Whitney-U test and Wilcoxon test. It is fair to use the Mann-Whitney-U test for testing the a-priori significances in Table 3. However, from my point of view, for the experimental setting that provided the data in Table 4, a repeated-measures-ANOVA is the only correct way to evaluate the data and the results from the Mann-Whitney-U and Wilcoxon test should NOT be reported. Instead, the p values from the rmANOVA including the interaction term along with the p values from Tukey’s post-hoc test should be stated in Table 4 and considered in the Discussion. As mentioned before, I do regard the reporting of the results from the latter tests as mandatory. I am pretty sure that using the multivariate testing won’t change the authors’ principal findings.

Abstract:

Line 23: Unclear what the authors mean with “it” here – the intervention? If so, please replace.

Author Response

Responses to Reviewer #1:                                                                                                           

  1. Shufan and colleagues have made amendments to their original manuscript on cost associated with health-promoting eating. The quality of the paper has largely improved, but two points, one major and one minor, still demand to be addressed before the manuscript finally can be recommended for publication.

The authors’ response to Points # 1 and #12 (as numbered by the authors) is confusing and contradictory. In Lines 151-158, it is still written that they applied two-group comparisons such as the Mann-Whitney-U test and Wilcoxon test. It is fair to use the Mann-Whitney-U test for testing the a-priori significances in Table 3. However, from my point of view, for the experimental setting that provided the data in Table 4, a repeated-measures-ANOVA is the only correct way to evaluate the data and the results from the Mann-Whitney-U and Wilcoxon test should NOT be reported. Instead, the p values from the rmANOVA including the interaction term along with the p values from Tukey’s post-hoc test should be stated in Table 4 and considered in the Discussion.

As mentioned before, I do regard the reporting of the results from the latter tests as mandatory. I am pretty sure that using the multivariate testing won’t change the authors’ principal findings.

Response: We adopted the statistical approach suggested by the reviewer and revised the manuscript accordingly. Specifically, based on our study design, satisfaction from various aspects of the menu served in each dining room was compared between two groups, The comparison was conducted in 2-time points, baseline, and 3 months afterward. We performed a repeated measure ANOVA for our satisfaction data including testing the interaction term as was requested by the reviewer. The relevant parts of the paper [Abstract, Methods, Results, and Discussion] were modified accordingly.

- The corresponding part in the Abstract was revised. It now reads: “While diners’ satisfaction decreased in the control group, it did not change in the intervention group following the intervention (p=0.018).

- The corresponding part in the Methods was revised. It now reads: “To examine the impact of the intervention on between-groups and within-group changes in diners’ satisfaction, we used a repeated measure ANOVA where the core variable of interest was the interaction between the effects of time (before vs. after the intervention) and group (study vs. control).”

- The corresponding parts in the Results section were revised accordingly. Table 4 was replaced by a revised Table 4 and the text is now: “Table 4 presents the mean (±SE) diners’ satisfaction with the catering service before and after the intervention. While there was a decrease in the average satisfaction with the catering service among the control group, no change was observed in the intervention group following the program (p=0.018). The significant between-groups differences following the intervention stemmed predominantly from the general level of satisfaction with the kitchen and dining room (p=0.037), the variety of dishes served (p=0.007), and the nutritional values of the dishes served (p=0.045).

- The corresponding sentence in the Discussion section were revised: “In our study, while there was a decrease in diners’ satisfaction in the control group, satisfaction with the catering service after the intervention did not change in the intervention group.

  1. Abstract: Line 23: Unclear what the authors mean with “it” here – the intervention? If so, please replace.

Response: Done.
